# In Vitro-Based Production of Virus-Tested Babaco (*Vasconcellea x heilbornii*, *syn. Carica pentagona*) in Ecuador: An Integrated Approach to an Endangered Crop

**DOI:** 10.3390/plants12132560

**Published:** 2023-07-06

**Authors:** Valeria Muñoz, Diana Curillo, Sebastián Gómez, Lissette Moreno-Peña, Denisse Peña, Eduardo Chica, Viviana Yánez, Eduardo Sánchez-Timm, Diego F. Quito-Avila

**Affiliations:** 1Centro de Investigaciones Biotecnológicas del Ecuador (CIBE), Escuela Superior Politécnica del Litoral, ESPOL, Guayaquil 090101, Ecuador; vkmunoz@espol.edu.ec (V.M.);; 2Facultad de Ciencias Agrícolas, Universidad de Cuenca, Campus Yanuncay, Av. October 12 and Diego de Tapia, Cuenca 010107, Ecuador; 3Facultad de Ingeniería y Ciencias Aplicadas, Carrera de Ingeniería en Agroindustrias, Universidad de las Américas, Campus UDLAPARK, Quito 170503, Ecuador; 4Facultad de Ciencias de la Vida, FCV, Escuela Superior Politécnica del Litoral, ESPOL, Guayaquil 090101, Ecuador

**Keywords:** babaco, virus, in vitro propagation, babaco certification

## Abstract

Babaco (*Vasconcellea x heilbornii*), a fruit-bearing vegetatively propagated crop native to Ecuador, is appreciated for its distinctive flavor and nutritional properties. The aim of this research was to determine a functional protocol for tissue culture propagation of virus-free babaco plants including in vitro establishment, multiplication, rooting, and acclimation. First, symptomless babaco plants from a single commercial nursery were analyzed for virus detection and cared for using different disinfection treatments in the greenhouse to reduce contamination during the in vitro establishing step, and three cytokinins, 6-(γ,γ-Dimethylallylamino) purine (2IP), 6-Benzylaminopurine (BAP), and Thidiazuron (TDZ), were used to determine the best hormone for multiplication. The best treatment for plant disinfection was the weekly application of copper sulfate at the greenhouse and a laboratory disinfection using ethanol (EtOH) (70%), Clorox (2%), and a solution of povidone iodine (2.5%), with an 80% survival during in vitro plant establishment. TDZ showed a better multiplication rate when compared with other hormones, and 70% of the rooted plants were successfully acclimated at the greenhouse. Generated plants were virus-free when tested against babaco mosaic virus (BabMV) and papaya ringspot virus (PRSV), two of the most important viruses that can affect babaco. An efficient protocol to produce virus-free babaco plants was elaborated with an integrated use of viral diagnostic tools to ensure the production of healthy start material to farmers.

## 1. Introduction

Babaco (*Vasconcellea x heilbornii*) is an interspecific hybrid resulting from natural crosses between different Vasconcellea species collectively known as mountain papayas [1]. The plant is native to subtropical southern areas of Ecuador, where it can be found in altitudes between ~1200 and 2200 m above sea level. It belongs to the *Caricaeae* family, having morphological and anatomical similarities with papaya (*Carica papaya*). Babaco fruits have an elongated pentagon-like shape with fresh flesh and a strong, stringent flavor which makes them suitable for juices, jams, and other processed products [2].

Babaco production represents a major income source for hundreds of families throughout subtropical areas of Ecuador. However, pathogen-related problems reduce yield and significantly shorten the production life of this native crop. Among several pathogens reported for this crop, systemic ones such as viruses constitute a major issue due to the reduction in yield, fruit damage, and shortened life span these pathogens cause to the plant [3]. Moreover, the vegetative propagation nature of babaco has had a significant impact on the accumulation and spread of viruses throughout commercial nurseries and in production fields, as this species produces seedless fruits [4].

As a typical vegetative propagated crop, commercial babaco production has two well-defined components. The first involves nursery operations in the southern provinces, near the hypothetical origin center of the plant; the second component involves field or greenhouse production operations, which take place in the northern provinces of the country. Nursery operators hunt for babaco plants growing under natural conditions for propagation by stem cutting without virus detection protocols. Therefore, a single virus-infected plant can result in, usually, 15 to 20 daughter infected plants. This practice has been carried out for decades, causing an exponential spread of virus-infected plants, followed by vector-mediated plant-to-plant transmission within the field [3,5].

Efforts to effectively propagate babaco through tissue culture have been unsuccessful. Since the babaco tissue used for in vitro culture comes from long vegetatively-propagated plants, it contains high populations of endophytic fungi and bacteria, which hamper the early phases of in vitro establishment. Hence, it has not yet been possible to establish a methodology that includes establishment, propagation, rooting, and acclimatization in greenhouses. Furthermore, the in vitro propagation of babaco must include virus testing and elimination, if necessary, to ensure virus-free, high-quality material for producers.

Along this line, our group has previously worked on the identification of several viruses from symptomatic and asymptomatic babaco plants in nurseries and production fields [3,6]. Thus far, out of the growing list of babaco-infecting viruses, papaya ringspot virus (PRSV) and babaco mosaic virus (BabMV) are particularly important in babaco due to their prevalence and the severe symptoms they can cause, including leaf mosaic, deformation, and necrosis [3]. Additional viruses such as babaco cryptic virus-1 (BabCV-1, a partiti-like virus) and babaco endogenous pararetrovirus (BabEPV, a hypothetical integrated pararetrovirus) have been reported in asymptomatic babaco [3]; however, their implication in babaco diseases when present in co-infection with other viruses remains to be determined. Hence, testing for both symptomatic and asymptomatic viruses must be implemented in virus-free babaco propagation programs.

The objective of this study was to develop an effective methodology for the in vitro propagation of babaco plants to generate high-quality propagation material for nurseries and producers.

## 2. Results and Discussion

### 2.1. Virus Detection

A total of 35% of asymptomatic babaco plants from the commercial nursery tested positive for BabMV, PRSV, or both, while 5% were positive for BabChV-1, BabNV-1, or BabMelV. BabVQ (an umbra-like virus) and BabCV-1 (a partiti-like virus) were present in all asymptomatic plants. However, previous work has shown that these two viruses (BabVQ and BabCV-1) do not induce discernible symptoms in babaco plants (Quito-Avila, D.F. unpublished). BabMV- or PRSV-infected plants kept under greenhouse conditions displayed symptoms after four or five weeks of testing, highlighting the importance of virus testing even in asymptomatic babaco plants at the nursery level.

### 2.2. Greenhouse Conditions and Sanitation Treatments for In Vitro Establishment

We used a combination of different antibiotics and fungicides at the greenhouse level. Streptomycin and Gentamicin are antibiotics from the aminoglycosides group, which prevent the onset of protein synthesis by binding to the 30S ribosomal subunits causing cell death [7]. Azoxystrobin and difenoconazole, both broad-spectrum systemic fungicides, were combined with contact fungicides such as Maneb and zineb, two dithiocarbamates, and copper sulfate [7]. With 66.7% explant survival, the best disinfection treatment was DT4, which involved the weekly application of copper sulfate as part of greenhouse sanitation and the immersion of the explants in a povidone solution prior to in vitro establishment. There are no previous studies indicating the use of povidone solution as a disinfecting agent in babaco. However, its effectiveness has been proven in other crops such as *Origanum vulgare*, *Dioscorea* spp., and *Buddleja incana.*

Camacho et al. [8] reported zero contamination applying povidone solution combined with a 1.5% NaClO solution in *Origanum vulgare*. On the other hand, Ramos et al. [9] obtained viable explants using a 3.5% povidone solution combined with a 2.5% NaClO solution in *Dioscorea* spp. Additionally, Jiménez et al. [10] mentioned the use of 1% of povidone solution with a mix of fungicides, antioxidants, a commercial fruit disinfectant, and active carbon for a successful in vitro establishment of *Buddleja incana*.

Guerrero et al. [11] reported 45% of viable explants using carbendazim (fungicide) in greenhouse sanitation of babaco plants, combined with a laboratory disinfection using 1.5% NaClO solution immersion for 10 min. A similar disinfection method was used in DT3, where a fungicide and antibiotic greenhouse application of plants combined with a 2% NaClO solution immersion for 10 min for in vitro disinfection was used, leading to a 57% explant survival.

When explant survival was analyzed by type of stem cut (e.g., apical, middle, or basal stem section), most of the basal segments presented low mean survival percentages. However, explants belonging to DT4 showed 80% survival overall (Table 1). Explants from treatments DT3 and DT6 showed 60% and 70%, respectively, survival for more than one type of explant.

### 2.3. Multiplication Rate

The use of different growth regulators was evaluated to improve the in vitro multiplication rate in babaco. The best treatment in terms of multiplication rate was MT4 with 4.01 shoots per plant, followed by MT3 with 2.03 shoots per plant (Figure 1).

These results differ from those obtained by Culqui et al. [12], where they report a multiplication rate of 4.60 shoots per plant using 0.5 mg L^−1^ of BAP in babaco explants. Additionally, Vélez-More et al. [13] indicated an average of four shoots per plant using BAP (2 uM) and ANA (0.54 uM) tested in somatic embryos of babaco.

Thidiazuron (TDZ) is a cytokinin commonly added to the multiplication culture media to accelerate organogenesis and plant regeneration. A previous study conducted in leaf explants of babaco reported positive results using TDZ as a high organogenesis promoter, showing 67% of regenerated tissue with 5 mg L^−1^ of TDZ [13]. Chérrez et al. [14] achieved a multiplication rate of 6.3 shoots per explant in papaya (*Carica papaya*) using a concentration of 2.5 mg L^−1^ of TDZ in multiplication culture media.

### 2.4. Plant Growth

For growth, the best height average of 1.95 cm was obtained using medium supplemented with 2 IP at 1 mg L^−1^ as part of MT2 (Figure 2). Vidal et al. [15] mentions that the hormone 2IP favored the greatest development of shoots because in this study they were able to obtain the highest growth (3.2 cm) with 2.5 mg L^−1^ of 2IP in *Carica papaya* shoots.

Despite having shown good results in multiplication rate, the treatments with BAP and TDZ did not obtain a good growth rate. In a study made in *Carica papaya*, an average of 1.54 cm of height was reported using 0.5 mg L^−1^ of BAP + 0.5 mg L^−1^ of IAA + 0.3 mg L^−1^ of AG3 [16]. Generally, the different segments had similar responses for each treatment, and therefore there were not statistically significant differences between the treatments and their viability. Finally, the shoots obtained in vitro were rooted with indole butyric acid (IBA) to be acclimatized in the greenhouse.

### 2.5. Acclimatization

A total of 66.7% of plants were successfully acclimatized in the greenhouse following the established protocol (Figure 3). There was a clear correlation between acclimation success and the conditions of the explants which included good root development, height greater than 1 cm, and good foliar development.

The maintenance of the environmental conditions and complying with the established times in the protocol also contributed to the correct acclimatization of the plants. When placed at an elevated temperature with direct exposure to the sun and a relative humidity under 90%, plants progressively died.

For in vitro propagated *Carica papaya*, different acclimation percentages have been reported, ranging from 40% to 96.5% [17,18,19], controlling variables similar to those used in this study such as relative humidity, substrate with MS salts, acclimatization time, and environmental conditions.

## 3. Materials and Methods

### 3.1. Plant Material and Virus Detection

Two-hundred symptomless babaco plants were purchased from a single commercial nursery in Paute, Azuay province (Ecuador). Plants were tested for BabMV, PRSV, babaco cheravirus-1 (BabChV1), babaco nepovirus-1 (BabNV1), and babaco meleira-like virus (BabMelV) as described in Cornejo Franco et al. [5] (primer list shown at Appendix A). Total RNA extraction and reverse-transcription (RT) were performed as described in Halgren et al. [20]. Briefly, 100 mg of leaf tissue was ground in lithium chloride-based extraction buffer, followed by protein precipitation using 6 M potassium acetate. RNA was pelleted using isopropanol and washed twice with Tris-based washing buffer mixed with 100% (1:1) ethanol in the presence of glass milk (silica). PCR was performed using GoTaq^®^ Green Master Mix (Promega, Madison, WI, USA) following the manufacturer’s instructions. Cycling parameters and specific primers were those described by Cornejo-Franco et al. [5].

Babaco plants negative for BabMV and PRSV, as well as other symptom-associated viruses such as babaco cheravirus-1 (BabChV1), babaco nepovirus 1 (BabNV1), and babaco meleira-like virus (BabMelV), were transferred to one-gallon plastic bags filled with a commercial substrate LM-18 (https://lambertpeatmoss.com/en/products/lm-18-germination-mix/) (accessed on 7 June 2023), grown, and kept under greenhouse conditions (25 °C and 12 h of light). LM-18 is a soil mixture that contains peat and fine perlite, limestone, and a wetting agent, which is commonly used for germination and the start-up of cuttings.

### 3.2. Greenhouse and Laboratory Sanitation Treatments before In Vitro Introduction

Virus-tested plants were randomly divided into nine groups (*n* = 10 each). A combination of disinfection treatments (DTs) both at the greenhouse and laboratory levels was applied to each group (Table 2). In the greenhouse, DTs were based on weekly applications of streptomycin, an antibiotic, and two broad-spectrum commercial fungicides, one based on 200 mg/L azoxystrobin and 125 mg L^−1^ difenoconazole, 2.5 g L^−1^ dithiocarbamates (Maneb and Zineb), and 2.5 g L^−1^ copper sulfate. After eight weeks, 5 cm long stem fragments were cut from the basal, medium, and apex sections. The cuttings were brought immediately to the laboratory and treated with solutions of 10% povidone-iodine and gentamicin (200 mg L^−1^). Table 2 shows the treatments and concentrations used for the sanitation phase at the greenhouse and laboratory levels.

After the corresponding treatment, the cuttings were placed in glass jars containing 30 mL of sterile Murashige and Scoog (MS) basal culture medium [21], with Gambor salts (SIGMA M0404, St. Louis, MO, USA), 30 g L^−1^ sucrose, 7 g L^−1^ agar, and a pH of 5.7; they were then kept in standard conditions with a photoperiod of 16:8 h of light/dark at 25 °C for 30 days.

### 3.3. Multiplication and Rooting

To select the optimal cytokinin for babaco propagation, three multiplication treatments (MTs) were established with three different hormones, respectively, in the MS culture medium: BAP, 2IP, and TDZ, in combination with 0.5 mg L^−1^ indole-3-acetic acid (IAA), shown in Table 3. After four weeks, the rate of multiplication and growth was recorded to continue with the propagation, with only two treatments showing the best multiplication rate. Plants were rooted in half-strength MS basal medium, supplemented with 1 mg/L IBA [16].

### 3.4. Acclimatization

The rooted plants from the rooting stage were carefully washed with distilled water to remove the remains of agar from the roots. Plantlets were transferred to plastic trays containing sterile LM-18 substrate supplemented with liquid MS salts without sucrose and 1 mL of dithiocarbamate- and copper sulfate-containing fungicide (2.5 mg mL^−1^ of distilled water). The trays were covered with plastic lids to maintain high humidity (>90%) and placed under a black Saran inner cover, with a photoperiod of 12 h of daylight with a temperature of 25 °C for 4 weeks. During the second week, plants were fertilized by direct irrigation of soil with a solution of liquid ½ MS salts supplemented with 2 mg/L IBA and 0.5 mg/LNAA, followed by a spray with distilled water, to evenly irrigate the plastic tray.

After four weeks, plants with new leaves were transferred to plastic cups containing LM-18 substrate moistened with liquid ½ MS salts and kept, under the same environmental conditions described above, for two months with the application of a weekly spray of distilled water. Once the acclimatation steps were finished, the percentage of viability and the survival rate were measured. With these results, the protocol for babaco propagation was completed and it can be downloaded in the Appendix A.

### 3.5. Statistical Analysis

The variables were subjected to variance analysis (ANOVA) and Tukey test to determine statistically significant differences among treatment means at 5% level of probability (*p* < 0.05) using R version 4.1.2 (1 November 2021)—“Bird Hippie” program.

## 4. Conclusions

Fungicides were effective in greenhouse sanitation at controlling the contamination of babaco plants. In general, disinfection treatments—without greenhouse sanitation—showed low percentages of viable explants during in vitro acclimatation. Povidone iodine proved to be an effective surface disinfection agent in cuttings; combined with the application of fungicides in the mother plant and together with disinfection with sodium hypochlorite, good results were obtained in the introduction and establishment of babaco cuttings. Because there is little information on the use of povidone iodine in this crop, it is recommended to carry out new experiments where other concentrations and other reagents such as the Plant Preservative Mixture (PPM) are tested.

TDZ was the most efficient phytohormone for the in vitro propagation of babaco, as it generated a large number of new shoots per plant. However, it caused a high swelling at the base of the outbreak, so it is recommended not to prolong its use or not to use amounts above 3 mg L^−1^. Due to the limited information on the use of TDZ in babaco, it is recommended to perform new tests with concentrations between 0 and 2 mg L^−1^.

BAP also proved efficient in the in vitro spread of babaco; however, other studies report a higher number of outbreaks generated with a higher amount compared with that used in this study. Therefore, it is proposed to carry out new tests with higher amounts of BAP for the generation of new outbreaks in future experiments.

In vitro propagation of babaco has not been efficient and not well-applied to provide producers with healthy start material. This study contributes to the development of efficient in vitro propagation protocols for the complete plant production process, including a pre-laboratory phase, to ensure a proper and successful tissue culture establishment, multiplication, and rooting of babaco explants. This work goes along with undergoing studies to determine the babaco virome and lay the foundation for the development of a certification program in Ecuador to produce healthy virus-tested babaco plants at the nursery level.

## Figures and Tables

**Figure 1 plants-12-02560-f001:**
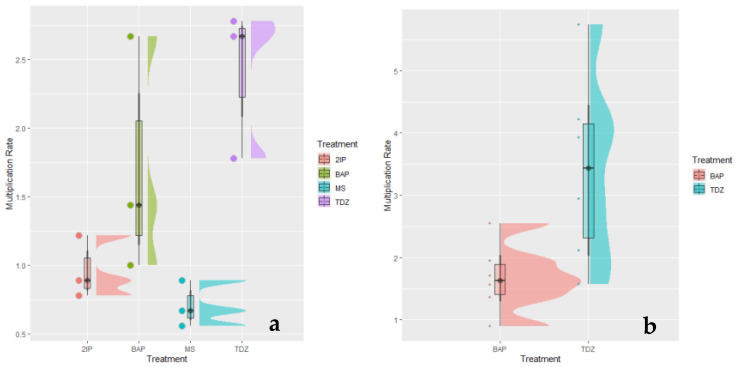
Multiplication rate of the established babaco (*Vasconcellea x heilbornii, syn. Carica pentagona)* plants. (**a**) Second subculture with four treatments 6-(γ,γ-Dimethylallylamino) purine (2IP), 6-Benzylaminopurine (BAP), MS (no hormone, and Thidiazuron (TDZ)) (*p* = 0.169). (**b**) Third subculture with the two selected best treatments (BAP and TDZ) (*p* = 0.0005).

**Figure 2 plants-12-02560-f002:**
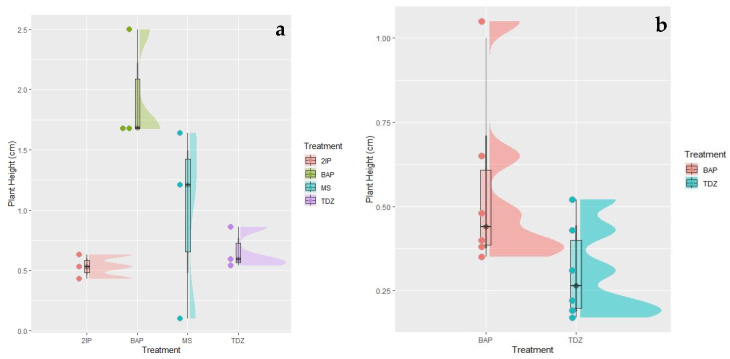
Plant height (cm) of the established *Vasconcellea x heilborni* plants. (**a**) Second subculture with four treatments 6-(γ,γ-Dimethylallylamino)purine (2IP), 6-Benzylaminopurine (BAP), MS (no hormone), and Thidiazuron (TDZ) (*p* = 0.02). (**b**) Third subculture with two selected best treatments (BAP and TDZ) (*p* = 0.01)).

**Figure 3 plants-12-02560-f003:**
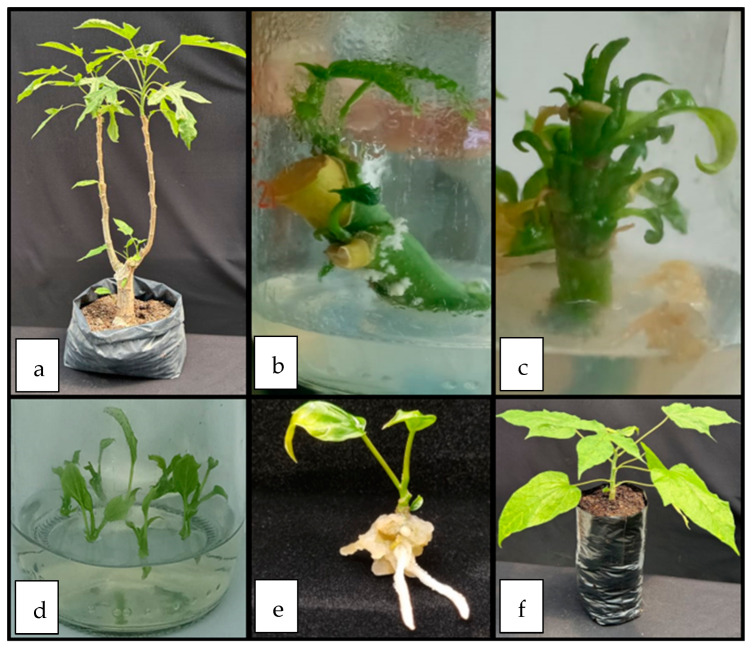
Propagation process of babaco plants (**a**) original “mother” plant grown at a greenhouse, (**b**) stem cuttings at the in vitro introduction process, (**c**) multiplication stage, (**d**,**e**) plant rooting, and (**f**) greenhouse plant acclimation.

**Table 1 plants-12-02560-t001:** Mean plant viability percentage through in vitro establishment of explants applied with different disinfection treatments (DTs).

Treatment	Percentage of Plant Viability	Percentage of Plant Viability by Segment Type		
		Apical	Middle	Basal
DT1	56.7%	50%	70%	60%
DT2	13.0%	0%	10%	10%
DT3	57.0%	60%	60%	40%
DT4	66.7%	80%	80%	80%
DT5	13.0%	10%	10%	10%
DT6	50.0%	70%	70%	10%
DT7	27.0%	60%	40%	10%
DT8	10.0%	10%	10%	0%
DT9	43.0%	60%	50%	50%

**Table 2 plants-12-02560-t002:** Sanitation and disinfection treatments, DTs, at greenhouse and laboratory levels.

	Greenhouse	Laboratory
**DT1**	Weekly application of 5 mL of AmistarTop (1 mL L^−1^) and 500 mL of streptomycin (200 mg L^−1^) per plant.	Disinfection with EtOH (70%) for 1 min and Clorox (2%) for 10 min, and immersion of the stem cuttings in a solution of povidone iodine (2.5%) for 5 min followed by overnight draining inside a sterile glass.
**DT2**	Weekly application of 5 mL of AmistarTop (1 mL L^−1^) and 500 mL of streptomycin (200 mg L^−1^) per plant.	Disinfection with EtOH (70%) for 1 min and Clorox (2%) for 10 min, and immersion of stem cuttings in a gentamicin solution (150 mg/L) for 5 min followed by overnight draining inside a sterile glass.
**DT3**	Weekly application of 5 mL of AmistarTop (1 mL L^−1^) and 500 mL of streptomycin (200 mg L^−1^) per plant.	Disinfection with EtOH (70%) for 1 min and Clorox (2%) for 10 min.
**DT4**	Weekly application of 5 mL of Cuprofix (2.5 g L^−1^) per plant.	Disinfection with EtOH (70%) for 1 min and Clorox (2%) for 10 min, and immersion of the stem cuttings in a solution of povidone iodine (2.5%) for 5 min followed by overnight draining inside a sterile glass.
**DT5**	Weekly application of 5 mL of Cuprofix (2.5 g L^−1^) per plant.	Disinfection with EtOH (70%) for 1 min and Clorox (2%) for 10 min, and immersion of stem cuttings in a gentamicin solution (150 mg/L) for 5 min followed by overnight draining inside a sterile glass.
**DT6**	Weekly application of 5 mL of Cuprofix (2.5 g L^−1^) per plant.	Disinfection with EtOH (70%) for 1 min and Clorox (2%) for 10 min.
**DT7**	No treatment.	Disinfection with EtOH (70%) for 1 min and Clorox (2%) for 10 min, and immersion of the stem cuttings in a solution of povidone iodine (2.5%) for 5 min followed by overnight draining inside a sterile glass.
**DT8**	No treatment.	Disinfection with EtOH (70%) for 1 min and Clorox (2%) for 10 min, and immersion of stem cuttings in a gentamicin solution (150 mg/L) for 5 min followed by overnight draining inside a sterile glass.
**DT9**	No treatment.	Disinfection with EtOH (70%) for 1 min and Clorox (2%) for 10 min.

**Table 3 plants-12-02560-t003:** Treatments with different combinations of phytohormones for multiplication.

	Treatments
**MT1**	MS
**MT2**	MS + 0.5 mg L^−1^ IAA + 1 mg L^−1^ 2IP
**MT3**	MS + 0.5 mg L^−1^ IAA + 1 mg L^−1^ BAP
**MT4**	MS + 0.5 mg L^−1^ IAA + 1 mg L^−1^ TDZ

## Data Availability

Data available on request from the corresponding author.

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
