# Peer review of "In Vitro-Based Production of Virus-Tested Babaco (Vasconcellea x heilbornii, syn. Carica pentagona) in Ecuador: An Integrated Approach to an Endangered Crop"

_plants, 2023, doi:10.3390/plants12132560_

Round 1

Reviewer 1 Report

The article seems to be original

The commercial cultivation protocol is the proof of the efficiency of the methodology.

I accept the article in its present form.

NOne

Author Response

Thank you for your time reviewing the manuscript.

Reviewer 2 Report

Manuscript of the message "In vitro-based production of virus-tested babaco (Vasconcellea x heilbornii, syn. Carica pentagona) in Ecuador: an integrated approach to an endangered crop" by Valeria Muñoz, Diana Curillo, Sebastián Gómez, Lissette Moreno-Peña, Denisse Peña, Eduardo Chica, Viviana Yánez, Eduardo Sánchez-Timm and Diego F. Quito-Ávila is a complete study aimed at obtaining a protocol for in vitro propagation of a rare cultivated plant. The manuscript contains all the necessary sections and is designed according to the rules.

There are some minor comments. I could not find a description of the harmfulness of viruses in the manuscript and what this harmfulness was expressed in: symptoms, reduced lifespan, drop in productivity or quality. It's not obvious, so an addition is required.

It is also not clear how harmful asymptomatic viruses are and, in this regard, how justified is their identification. This requires the disclosure of the topic.

The adaptation protocol is described, but no photographs are given. In addition, it is not clear to me whether the roots that were in the agar really contributed to survival, often this is not the case.

After adding explanations, the message can be published.

Author Response

  1. The description of harmfulness of virus has been included (also asymptomatic).
  2. A picture shows the adapted plant and the protocol describes the proccess, and no experiment was performed for rooting, and we do not discusse the usefulness of the roots on agar, only media rooted plants were transferred to the soil mixture. As you mentioned, usually agar roots are lost in the adaptation proccess, but this is not disscused in the present work. Thank you for your kind contributions, hope to cover all of them. You can see all the corrections in the uploaded document.
  3. A table with the primers names and sequences has been included as supplementary material.
  4. Concept of figure 1 and 2. Multiplication rate was calculated on different cytokinins treatments (figure 1), the best treatments were subcultured and multiplication rate were calculated again (figure 2). Violin figures from R were used to present the data, because they include data dispersion.
  5. There is no lack of statical analysis, all the analyses mentioned on materials and methods were performed, including one way ANOVA, and PostHoc Tuckey, to validate statistical significance and grouping diferentiation.

Reviewer 3 Report

Dear Editor,

I extend my sincere appreciation for affording me the opportunity to review the paper titled “In vitro-based production of virus-tested babaco (Vasconcellea 2 x heilbornii, syn. Carica pentagona) in Ecuador: an integrated approach to an endangered crop” by Muñoz et al. In my opinion, the paper is suitable for the special issue, however, it needs Major corrections as follows:

-        The abstract needs to be rewritten. The introduction sentence of the abstract is not suitable, its better if the sentence will be replaced with a sentence about the plant. It is good if the authors explain the material and method clearly and present the quantitative data in the results section.

-        Authors mentioned that they purchased the plants from a commercial market, however, how are they sure if the plants are from the same cultivar, size, age, resource, etc?

-        Line 188, please mention the name of viruses which you have tested the plants for their identification

-        Lines 188 and 190, please follow the journal format and style for the reference inside the text.

-        Line 195; “described by [5].” Please follow the journal format, and write the name of the first author.

-        Line 141, reference 14, please double-check the paper entirely. 

-        Please mention the name or sequence of specific primer.

-        I believe that Table 1 should be replaced under the relevant section in the material and methods section

-        Please explain the concept of figure 1 and 2

-        There is a lack of statical analysis, authors should run a statical analysis, explain the ANOVA result and mean of comparison, therefore, based on the mentioned results, explain their findings

-        The pictures don’t have a scale bar

-        There are so many typos, for example, make a space between figure and 3 or line 237; 0.5 mg/L ANA.

-        The number of references is not enough and shows that the literature and discussion are not strong

Should be improved

Author Response

  1. This crop is multiplied using cutting techniques, it is clonally propagated in plant nurseries. They were acquired as producers would do, there is no record of the plants. So they were purchased, adapted to greenhouse controlled conditions, pruned and disinfected as the manuscript describes. 
  2. Abbreviations are corrected.
  3. IBA concentration was decided using reference [16], it has been referenced already, thank you for making us noticing it.
  4. As the manuscript explains, different cytokinins were used, to analyze the best one for this particular crop. There is no previews reference of a complete multiplication protocol for babaco. Further analysis can be performed as recommended in future research.
  5. Abbreviation typos has been corrected.
  6. For a short communication, reference number is appropriated. Thank you for your concern in this matter.

Reviewer 4 Report

In vitro-based production of virus-tested babaco (Vasconcellea  x heilbornii, syn. Carica pentagona) in Ecuador: an integrated approach to an endangered crop

I suggest changing the title to be:

In vitro-based propagation of virus-free Babaco (Vasconcellea  x heilbornii, syn. Carica pentagona) in Ecuador: an integrated approach for commercial production to an endangered crop

The aim of this MS is to develop an efficient protocol to produce virus free babaco plant materials for farmers and nurseries

Material and Methods:

Line 210-215: How old are the plants when you take the explants (stem fragments) for sanitation then establishment phases   

The full name of plant growth regulators and its abbreviation (as 6-Υ,Υ-dimethylallylamine purine (2IP), should be written only when you mention them in the first time after that you have to use the abbreviation only.

" Plants were rooted in half strength MS, supplemented with 1 mg/L IBA" why did you use this treatment for rooting is it reported before? please explain and cite the ref if it is found 

Table 4:

You examine just four treatments for multiplication; I think it is not enough

What do you mean MS you have to write MS basal medium (without PGR, I mean)

Correct AIA to IAA in all MS

Why did you use 0,5 mg/L IAA  is it the optimal concentration for Babaco in vitro multiplication in combination with CKs, if yes cite the reference and why did you use 1 mg L-1 from each cytokinin, the same or mention the reason

0,5 mg/L IAA  replace it by 0.5 mg L-1  and correct all concentrations mentioned in the article

The title of table 4 is not correct, because it contains the treatment of shoot multiplication only, while rooting was one treatment (half strength MS, supplemented with 1 mg/L IBA) was applied for all multiplicated shoots

I think it is better to change the title into:

Different cytokinins supplemented to MS medium in combination with 0.5 mg L-1 IAA for multiplication of Babaco

Line 237: 0.5 mg/L ANA, you means NAA, please correct in all MS

Results and discussion:

Table (1) should be put in Material and Methods section, and show your obtained results by using the key of treatments as DT4

So, Table 2 will be the first table in your MS. Please consider the order of tables where table 2 will be table (1) and so on and refer to the correct number of table in the text

The foots of result tables (for example line 119) in the first table in the results section should contain the statistical analysis used in brief

Line 115:  table 1 not 2

Line 123-124: The sentence "The aseptic shoots obtained from the establishment stage were transferred to four different media culture described in Table 2 " . should be transferred to M and M section not here in the results

Table 3:  The treatments MT3 and MT4 is repeated twice with different results, please correct

What do you mean stage 1 and 2 I am really confused from the showing results

I couldn’t understand the figures 1 and 2

What do you mean " growth rate"  I think you mean the plant height

You presented your results of multiplication rate and plant height in table 3 , why you show them in figures I think one clear method is enough

Minor Revision 

 Extensive editing of English language required as the authors not native for English language 

Author Response

  • Typos has been corrected as suggested.
  • Table 1 has been relocated.
  • Teble 2 was renamed as table 1.
  • For the percentage results, now its written that those are mean percentages.
  • Table 3 (now table 2) has been removed as suggested, because same results were shown in two different ways. Other tables were renamed.
  • Figure 1 and 2 are figures showing growth and multiplication rates. The graphics are vioilin type from R analysis, which shows data dispersion in the same graphic.
  • Groth rate has been changed to plant height.